# Long-Chain Polyunsaturated Fatty Acids Are Associated with Blood Pressure and Hypertension over 10-Years in Black South African Adults Undergoing Nutritional Transition

**DOI:** 10.3390/foods8090394

**Published:** 2019-09-06

**Authors:** Manja M. Zec, Aletta E. Schutte, Cristian Ricci, Jeannine Baumgartner, Iolanthe M. Kruger, Cornelius M. Smuts

**Affiliations:** 1Centre of Excellence for Nutrition, North-West University, 2520 Potchefstroom, South Africa; 2Centre of Excellence in Nutrition and Metabolism, Institute for Medical Research, University of Belgrade, 11000 Belgrade, Serbia; 3Hypertension in Africa Research Team (HART), North-West University, 2520 Potchefstroom, South Africa; 4Medical Research Council Unit: Hypertension and Cardiovascular Disease, North-West University, 2520 Potchefstroom, South Africa; 5Africa Unit for Transdisciplinary Health Research (AUTHeR), North-West University, 2520 Potchefstroom, South Africa

**Keywords:** black Africans, blood pressure, hypertension, long-chain polyunsaturated fatty acids, longitudinal study, nutritional transition, PUFA

## Abstract

Nutritional transition in Africa is linked with increased blood pressure (BP). We examined 10-year fatty acid status and longitudinal associations between individual long-chain polyunsaturated fatty acids (PUFA), BP and status of hypertension (≥140/90 mmHg and/or medication use) in black South Africans. We included 300 adults (>30 years) participating in the Prospective Urban Rural Epidemiology study, and analysed data from three consecutive examinations (2005, 2010 and 2015 study years). Fatty acids in plasma phospholipids were analysed by gas chromatography-mass spectrometry. We applied sequential linear mixed models for continuous outcomes and generalized mixed models for the hypertension outcome, in the complete sample and separately in urban and rural subjects. Mean baseline systolic/diastolic BP was 137/89 mmHg. Ten-year hypertension status increased among rural (48.6% to 68.6%, *p* = 0.001) and tended to decrease among urban subjects (67.5% to 61.9%, *p* = 0.253). Regardless of urbanisation, *n*-6 PUFA increased and eicosapentaenoic acid (EPA, C20:5 *n*-3) decreased over the 10-years. Subjects in the highest tertile of arachidonic acid (C20:4 *n*-6) had 3.81 mmHg lower systolic (95% confidence interval (CI): −7.07, −0.54) and 3.82 mmHg lower diastolic BP (DBP) (95% CI: −5.70, −1.95) compared to the reference tertile, irrespective of lifestyle and clinical confounders. Similarly, osbond acid (C22:5 *n*-6) was inversely associated with DBP. Over the 10-years, subjects in the highest EPA tertile presented with +2.92 and +1.94 mmHg higher SBP and DBP, respectively, and with 1.46 higher odds of being hypertensive. In black South African adults, individual plasma *n*-6 PUFA were inversely associated with BP, whereas EPA was adversely associated with hypertension, supporting implementation of dietary fat quality in national cardiovascular primary prevention strategies.

## 1. Introduction

Urbanisation in sub-Saharan Africa followed by increased consumption of energy-dense food [1], is linked with an increase in cardiovascular disease, obesity and diabetes [2], and the highest prevalence of mean blood pressure (BP) since 1980 [3]. Among black South African adults (>30 years) participating in a large-scale Prospective Urban Rural Epidemiology (PURE) study, a five-year increase in hypertension rate has been reported [4]. Nutritional interventions remain a cost-effective approach in suppressing the hypertension burden in the population. Baseline data from the PURE study indicate low total fat and omega-3 (*n*-3) intake in black South Africans [5]. In the same study, dietary *n*-3 long-chain polyunsaturated fatty acids (PUFA) were associated with serum lipids. Dietary eicosapentaenoic acid (EPA; C20:5 *n*-3) was associated with dyslipidemia and docosahexaenoic acid (DHA; C22:6 *n*-3) with favourable lipid status in the population [5]. These results indicate a unique metabolic profile in black South Africans related to fat catabolism and a specific role of individual fatty acids in cardiometabolic function.

Long-chain *n*-3 PUFA from marine foods demonstrate BP-lowering effects [6]. Clinical studies show that long-chain *n*-3 PUFA consumption diminishes the risk of cardiac death, potentially through regulation of triglycerides, heart rate and BP [7]. Measurement of the intake remains a challenge, since questionnaires are imprecise in differentiating intake of individual long-chain fatty acids. Self-reported information from dietary questionnaires is further limited by recall bias and participants’ non-compliance to fat-intake related questions [8]. Fatty acids in plasma phospholipids are however reliable biomarkers reflecting fat intake of the preceding 6-8 weeks [9]. Dietary fatty acids are endogenously catalysed by desaturase-5 and desaturase-6 enzymes encoded by FADS1 and FADS2 genes, respectively. The conversion results in the formation of long-chain PUFA with diverse physiological functions. Plasma fatty acids are of raising importance as prognostic biomarkers of cardiovascular disease [10]. Data from the Women’s Health Initiative study show inverse association between *n*-3 group and coronary heart disease risk in post-menopausal women [11]. A recent review underlined the importance of individual circulating fatty acids with regards to total and cause-specific mortality, type 2 diabetes mellitus and cardiometabolic indices [10]. Plasma fatty acids have been associated with BP [12,13,14], and observational data suggest protective associations of individual circulating *n*-3 long-chain PUFA with BP [15,16]. In middle-aged and elderly Chinese community dwellers, serum patterns presenting with high DHA levels were inversely associated with BP [12] and hypertension status [17]. Recent data from the PURE study showed association between plasma phospholipid fatty acid patterns and obesity and metabolic syndrome in black South African adults [18], however the link with vascular function has not yet been examined in the population.

Therefore, the objective of this longitudinal study was to evaluate the relationship between fat intake and BP in black South Africans. We measured and reported fatty acids in plasma phospholipids over 10 years, in a sample of black South Africans participating in the PURE and residing in rapidly urbanizing areas. To address the study objective, we examined the associations between individual long-chain *n*-3 and *n*-6 PUFA with BP and hypertension status over the 10-years. We also evaluated the 10-year associations separately in subjects residing in urban and rural areas.

## 2. Materials and Methods 

### 2.1. Study Design and Selection of Study Sample

This study formed part of the South African cohort of the PURE study, an international study investigating health implications linked with urbanisation in low-, middle- and high-income countries [19]. The cohort included 2010 (1260 women and 750 men at baseline) randomly selected black adults (>30 years), from urban and rural areas of the North West Province, without use of chronic medication and/or any self-reported acute illness. Permission for the study was obtained from the provincial Department of Health. Trained fieldworkers fluent in both English and Setswana conveyed all information. All subjects voluntarily gave written informed consent for the participation in 2005, continuous consent throughout the study and again in 2010 and 2015. The study protocol adhered to the 1983 Declaration of Helsinki and was approved by the Health Research Ethics Committee of the Faculty of Health Sciences at the North-West University (Potchefstroom campus). Privacy and confidentiality were ensured during the data-gathering process, data and sample storage and management. 

For the purposes of this study and analysis we applied a repeated-measures design and included data and assessments from 2005 (baseline), 2010 and 2015 (follow-up). A sub-cohort of 711 subjects were randomly selected at 2010 (Appendix A
Appendix A) and fatty acid analysed accordingly. Due to the loss to follow-up in the 2015 study year, we ended up with 300 complete sets of samples across the 3 study years, inclusive of fatty acid profiles and vascular outcomes. The 300 corresponding subjects were thus included in the longitudinal analysis.

### 2.2. Clinical and Biological Measurements

Fasting blood samples were collected from the antecubital vein with a sterile winged infusion set and were with minimal stasis. The samples were collected by a registered nurse and stored at −80 °C. In rural areas the samples were stored at −18 °C up to 5 days; afterwards transported to the laboratory facility and stored at −80 °C until analysis. Plasma phospholipid fatty acid composition was analysed as described previously [20]. Briefly, lipids were extracted with chloroform: methanol (2:1 v/v) from thawed ethylenediaminetetraacetic acid-plasma samples according to the modified Folch method [21]. The phospholipid fatty acid fractions were subsequently isolated by thin layer chromatography, further transmethylated to fatty acid methyl esters, and analysed by quadrupole gas chromatography electron ionization mass spectrometry using an Agilent Technologies 7890 A GC system [20]. Levels of each individual fatty acid were expressed as a percentage of the total phospholipid fatty acid pool in plasma. To examine longitudinal associations with vascular function, we used data for the long-chain *n*-3 fatty acids: EPA, DHA and docosapentaenoic acid (C22:5 *n*-3); and *n*-6 fatty acids: dihomo-γ-linolenic acid (DGLA, C20:4 *n*-6), arachidonic acid (AA, C20:4 *n*-6), adrenic acid (C22:4 *n*-6), and docosapentaenoic acid (osbond acid, C22:5 *n*-6). 

Brachial BP was measured in duplicate in a sitting position by using a validated OMRON device (Omron Healthcare, Kyoto, Japan) after subjects rested for 10 min, as reported elsewhere [4]. To be categorized as hypertensive, the participants had to exceed either SBP (140) or DBP (90) or both thresholds (or had to use antihypertensive medication) [22]. The PURE-standardized demographic, socio-economic and lifestyle questionnaires were interviewer administered [19]. Education was confirmed if any formal education was present. Quantitative FFQ and the physical activity index questionnaire previously developed and validated for South Africans were used [23,24]. The FFQ was conducted in the morning on the study visit day. Study participants were provided with a list of food items (food or drinks) and were asked how often they had consumed specific foods or drinks on average in the preceding year. Assessment of height, weight, waist circumference, serum lipids and other biochemical measurements were described previously [4,5].

### 2.3. Statistical Analyses

Statistical analysis was performed using SAS 9.4 (SAS Institute Inc, Cary, NC, USA). Continuous variables were checked for the distribution by visual inspection of histogram and skewness. Normal, non-normal and categorical data are presented as mean ± SD, median [25th, 75th percentile] and percentages, respectively. Baseline between-subject differences across urbanisation areas were tested by independent *t*-test and Mann Whitney test, for normal and non-normal data, respectively. Between-subject differences across the three study years were analysed by general linear model. Pearson’s correlation coefficients were computed to evaluate the relationship between plasma long-chain phospholipid PUFA, *n*-3 intake (cumulative intake of dietary α-linolenic acid (ALA, C18:3 *n*-3), EPA and DHA) and marine fatty acid intake (cumulative intake of EPA and DHA).

We evaluated 10-year associations between individual *n*-3 and *n*-6 long-chain PUFA (exposures) and the outcomes, by inclusion of data for the three study points. We applied linear mixed models for continuous outcomes (SBP and DBP) and generalized mixed models for the outcome of hypertension, with individual exposure fatty acids included as tertiles of the plasma phospholipid content. Sequential regression-based models were applied: Model 1 with fatty acid predictors controlled for age, gender and level of urbanisation (urban or rural); Model 2 further controlled for lifestyle confounders, including level of education (no education or any kind of formal education), self-reported use of tobacco (current, former or never used), use of hypertension medication (yes or no), body mass index, physical activity index and dietary intake of alcohol (g per day). The urbanisation status was treated as random factor, and repeated measures design was accounted for by use of adequate syntax within the procedures. We performed a prespecified subgroup analysis, stratified by urbanisation status (rural and urban areas). The level of significance was set at 0.05 (2-tailed).

### 2.4. Sensitivity Analyses

We further tested whether multivariable-adjusted associations were independent on dietary and fat intake in three consecutive steps: adjusting for total energy intake, following total fat and carbohydrate intake, and lastly ratio of monounsaturated to saturated fat intake, and soluble fiber intake, latter known to be protective towards vascular health [25,26]. We also tested whether our associations survived upon adjustment for potential effect mediators linked with BP, including total cholesterol, triglycerides, fasting glucose, and γ-glutamyl transferase reported to be associated with hypertension in this population [4].

## 3. Results

### 3.1. Baseline Characteristics of the 300 Rural and Urban Black South Africans

The study sample included 300 black South Africans (mean age = 53.12 ± 9.83), out of which 91 were men, 140 rural residents (46.7%), and 42.3% overweight subjects, mostly women (53.6% and 16.5% women and men who are overweight, respectively, *p* < 0.0001). In total, 19.7% and 23% subjects had elevated total cholesterol and triglycerides, respectively. Only 2.3% of participants were either diagnosed with type 2 diabetes or actively taking anti-diabetic medications. 

Urban subjects presented with higher hypertension prevalence and had higher SBP and DBP, then rural residents. The intake of total energy, total carbohydrates, total fat, and specific fat groups were higher among urban subjects (Table 1).

### 3.2. Ten-Year Changes in Blood Pressure and Status of Hypertension

Within all 300 subjects, we observed a non-significant increase in hypertension rate (58.7%, 61.3% and 65% in 2005, 2010 and 2015, respectively; *p* = 0.210). There was a significant increase in the hypertension rate in rural residents (48.6%, 51.4% and 68.6% in 2005, 2010 and 2015, respectively; *p* = 0.001), and a non-significant decrease within urban dwellers (67.5%, 70% and 61.9% in 2005, 2010 and 2015, respectively; *p* = 0.253). At baseline and in 2010 there were significantly more hypertensive subjects in urban areas, with no differences in 2015. SBP and DBP significantly decreased across the 10-years in urban areas (Appendix A
Appendix A).

### 3.3. Ten-Year Changes in Long-Chain Plasma Phospholipid Fatty Acids

There was a significant decrease in γ-linolenic acid. Long-chain *n*-6 PUFA (DGLA, AA, adrenic and osbond acid) increased and long-chain *n*-3 (EPA and DHA) decreased across the 10 years (Table 2). Ten-year fatty acid status across urbanisation areas is presented in Appendix A
Appendix A. Regardless of urbanisation level, we observed increases in DGLA, AA and osbond acid. In urbans only, adrenic acid increased and EPA and DHA decreased over the 10-years. In rural subjects, docosapentaenoic acid and DHA content increased and EPA tended to decrease.

### 3.4. Relationship Between Dietary Intake of N-3 Fatty Acids and Long-Chain Fatty Acids in Plasma Phospholipids

A HeatMap of Pearson correlations among baseline intake of *n*-3 fatty acids and long-chain plasma phospholipid fatty acids is presented in Figure 1. 

Cumulative *n*-3 intake was correlated with status of EPA and DHA, and negatively correlated with *n*-6 long-chain PUFA: AA (*r* = −0.113, *p* = 0.052), adrenic acid (*r* = −0.280, *p* = < 0.0001) and osbond acid (*r* = −0.198, *p* = 0.001) (Appendix A
Appendix A). Cumulative intake of preformed EPA and DHA did not correlate with status of any long-chain *n*-3 PUFA, yet negatively correlated with long-chain *n*-6 PUFA: AA (*r* = −0.136, *p* = 0.018), adrenic acid (*r* = −0.126, *p* = 0.030) and osbond acid (*r* = −0.180, *p* = 0.002).

Fatty acids in plasma phospholipids within either the *n*-3 or *n*-6 group were correlated among each other. Adrenic acid inversely correlated with EPA and DHA. Osbond acid inversely correlated with EPA, while the inverse relationship with DPA and DHA did not reach statistical significance (Appendix A).

### 3.5. Ten-Year Associations between Long-Chain Plasma Phospholipid Fatty Acids with Blood Pressure and Hypertension in Black South African Adults

#### 3.5.1. Associations between *N*-3 Long-Chain Fatty Acids and Blood Pressure

Subjects in the highest tertile of plasma EPA content over the 10-years had 1.94 mmHg higher DBP in comparison with subjects in the lowest (multivariable β for T3 vs. T1: 1.94 (95% CI: 0.01, 3.87)), regardless of age, gender, BMI, educational background, intake of alcohol, smoking status, level of physical activity and use of hypertension medication (Table 3). 

Urban subjects within the highest EPA content were with higher SBP (Appendix A). DHA was inversely associated with DBP in rural dwellers (multivariable β for T3 vs. T1: −3.91 (95% CI: −7.04, −0.78).

#### 3.5.2. Associations between *N*-6 Long-Chain Fatty Acids and Blood Pressure

Across the 10 years AA was inversely associated with SBP and DBP (Table 3). Subjects in the highest tertile were with 3.81 and 3.82 mmHg lower SBP and DBP, respectively, in comparison with subjects within the reference tertile (multivariable β for T3 vs. T1: −3.81 (95% CI: −7.07, −0.54) for SBP and −3.82 (95% CI: −5.70, −1.95) for DBP). Osbond acid was inversely associated with DBP (Table 3). Subjects in the highest tertile had 3.71 mmHg lower DBP in comparison with reference tertile (multivariable β for T3 vs. T1: −3.71 (95% CI: −5.73, −1.70)). 

The inverse associations remained significant in urban residents for both AA and osbond acid, and osbond acid was also inversely associated with SBP (Appendix A). In urban dwellers DGLA was inversely associated with DBP.

#### 3.5.3. Associations between Long-Chain Plasma Fatty Acids and Status of Hypertension

Plasma phospholipid fatty acids were not associated with 10-years status of hypertension in the 300 black South African adults, except for EPA (Figure 2). Subjects in the highest tertile of EPA content were with 1.46 higher odds of being hypertensive across the 10-years, in comparison with those in the reference tertile (multivariable OR for T3 vs. T1: 1.46 (95% CI: 1.03, 2.08)) (Appendix A). Adverse relationship of DGLA was lost upon controlling for potential confounders.

EPA remained adversely associated with 10-year hypertension status only in rural subjects within the highest tertile of the content. Furthermore, DGLA and osbond acid were adversely associated upon controlling for confounders known to influence the status. No associations were seen among urban dwellers.

#### 3.5.4. Sensitivity Analyses

Observed 10-year associations with BP and hypertension status remained consistent upon sensitivity analyses evaluating contribution of dietary intake affecting fat metabolism, and serum biomarkers.

## 4. Discussion

Our study showed that in black middle-aged and elderly South Africans living in rapidly urbanizing areas, individual long-chain plasma phospholipid PUFA were associated with BP across 10 years. The *n*-6 fatty acids were protectively associated with office SBP and DBP, while subjects with the highest EPA content presented with higher DBP. The relationships were independent of age, gender, BMI, educational background, intake of alcohol, smoking status, level of physical activity, use of hypertension medication, total energy and intake of fat, and glucolipid biomarkers. Observed relationships between individual PUFA and vascular health confer the role of dietary fat quality in tailoring population-specific nutritional policies in black South Africans.

In our study EPA was adversely associated with 10-years status of hypertension. Previous studies suggest favourable associations of EPA intake with vascular function [7,27,28] and cardiovascular events [29]. In a prospective study among 1477 adult community dwellers, subjects in the highest quartile of erythrocyte EPA content had significantly lower SBP and DBP across 3 years [30]. However, the latter study included fatty acid biomarkers measured at single time-point, while our study considered time-dependent variations in the PUFA content by inclusion of the data from 3 consecutive examinations across the 10-years. Herein observed adverse EPA associations might be attributed to the aging of participants, an epidemiological context associated with increase in BP. EPA is a precursor of prostaglandins with limited vasodilatory properties and its physiological function might be outweighed by the natural course of aging. Further on, associations reflecting absolute changes with incremental EPA increase were relatively small. Subjects in the highest EPA tertile presented with only +2.92 and +1.94 mmHg higher SBP and DBP over the 10-years, respectively, in a multivariable-adjusted model. Of importance, baseline mean SBP/DBP was already higher (137/89 mmHg) and is with expected increasing trend over time due to aging, altogether potentially contributing to the observed 1.46 higher odds of being hypertensive with incremental EPA increase.

Our results should be interpreted in context of a population free of acute or chronic illnesses and residing in rapidly urbanizing areas. We showed raising hypertension prevalence across the 10-years, significant in rural areas. In urban dwellers we observed a non-significant decrease in hypertension rate, partly due to 10-years decline in both SBP and DBP of approximately 7 mmHg. Notably, 19.3% rural and 17.5% urban dwellers used hypertension medication at baseline. The number dramatically increased across the 10-years resulting in 35.7% and 33.1% of the respective subjects on medication in 2015, partly because study participants diagnosed with baseline high BP were instructed to their local clinics. Compliance with therapeutic protocols might be more prominent among urbans with readily available healthcare, resulting in a stabilization of hypertension prevalence across the 10-years. In our study, long-chain PUFA were not associated with status of hypertension across the 10-years, except for EPA being adversely related. Increased medication use might have masked the associations, due to the interaction with lipid metabolism [31]. The large-scale Atherosclerosis Risk in Communities study previously showed protective associations between total PUFA cholesterol ester content and 6-years prevalent and incident hypertension, with individual EPA and AA exhibiting adverse associations [32]. Overall, our results remain inconclusive on the association between fatty acids and hypertension status in the black South Africans, and larger cohorts should confirm the relationship.

The metabolic context of our results is of consideration. Within the sample of black South Africans, we found unusually high levels of long-chain PUFA in plasma phospholipids. Previously reported levels of serum AA were higher in African Americans with diabetes or metabolic syndrome, in comparison with their counterparts of European ancestry [33]. Still the levels were substantially lower (9.8 ± 1.9%) [33], in comparison with our study (mean range across the 10 years: 13.57–18.13%). In a larger population of Chinese subjects of similar age group as our participants, percentage of AA in total serum content was 6.02 ± 1.61 [12]. Also, in our study 10-years mean plasma phospholipid content of DHA exceeded 3.5%, which is above 2.5–3.4% previously reported in healthy populations [12,34,35,36]. Higher levels of long-chain AA and DHA observed herein might result from marked desaturase-6 activity. Observed DHA content is of special importance as only up to 1% of dietary ALA is endogenously converted to DHA [37] and our population had substantially low *n*-3 intake at baseline (year of 2005) [5] (median of 33 to 61 mg EPA +52 to 109 mg DHA below recommendations by FAO [38]). According to 2004 International Society for the Study of Fatty Acids and Lipids expert opinion, recommended combined EPA + DHA intake in general population should be at least 500 mg daily, conferring substantially low intake of the fatty acids in our subjects. A low fat, high carbohydrate diet is reported in other urbanizing populations [39] and is associated with augmented fatty acid synthesis [40]. We thus speculate that restricted intake of *n*-3 rich food in the black South Africans might be a conditional metabolic factor enhancing desaturase activity towards physiologically active long-chain plasma products, including AA and DHA. Notably, in our study baseline intake of marine PUFA was not correlated with its plasma phospholipid status. Previous results in 1834 Chinese community dwellers demonstrated strong correlation among erythrocyte long-chain *n*-3 content and their dietary counterparts [30]. However, when we evaluated *n*-3 intake as sum of preformed EPA, DHA and plant-originated ALA we observed a direct correlation with status of EPA (*r* = 0.138, *p* = 0.017) and DHA (*r* = 0.218, *p* = 0.000). The latter suggests that in our subjects, dietary ALA is pronouncedly converted towards plasma long-chain products by activity of desaturase enzymes. Previous reports indicate specific FADS genetic make-up in populations of African descent. Results from the Diabetes Heart Study showed that 81% of African Americans are carriers of FADS rs174537 variant [33], associated with AA, eicosadienoic acid and EPA levels [41]. We suggest that historically low intake of *n*-3 PUFA in the population of black South Africans is coupled with genetically-regulated higher metabolic conversion towards AA and DHA. 

We showed inverse associations of AA with BP across the 10-years. Observational studies found plasma AA to be protectively associated with coronary heart disease [42,43] and type 2 diabetes risk [44]. AA is a precursor of eicosanoids with pro-inflammatory properties and vasomodulatory function [45]. AA is also a precursor of epoxydes with anti-vasodilatory function, mediated by soluble epoxyde hydrolase [46]. A favourable balance between *n*-3 and *n*-6 intake potentiates production of vasodilatory eiocosanoids from AA and decreases BP [45,46]. Herein observed protective relationship of AA might be due to metabolic adaptation conditional to a historically low *n*-3 long-chain PUFA intake. The associations of AA were prominent within urban dwellers, potentially due to the interaction with micronutrient intake, such as magnesium known to influence desaturase-6 function [47]. In a previous cross-sectional study of 2447 middle-aged and older Chinese community dwellers, AA exhibited neutral associations with BP, but study subjects in the highest tertile of serum DHA had significantly lower SBP and DBP in comparison with those in the lowest [12]. Although there was an inverse trend, DHA was not significantly associated with BP, potentially due to limited size of our study sample. It is possible that in our population with inherently low *n*-3 PUFA intake, extensive conversion to DHA underpins its incorporation in phospholipid cellular bilayers for non-vascular beneficial effects. Prospective analysis among 381 healthy, middle-aged and elderly subjects participating in the Kuopio Ischemic Heart Disease Risk Factor study also failed to demonstrate associations between individual long-chain *n*-3 serum PUFA and BP over 10 years [48]. 

We observed protective associations of osbond acid with BP. Dietary contribution to osbond acid status is negligible and its physiological role is due to metabolic conversion. To our knowledge, no previous study reported associations of osbond acid with clinical outcomes. In our study, the 10-year increase in *n*-6 AA and osbond acid were related to clinically relevant 3–4 mmHg lower BP for subjects within the highest tertile of the PUFA content. The protective associations might reflect pronounced utilization of the *n*-6 long-chain products for physiological function in this population with restricted *n*-3 intake. The suggestion to increase *n*-6 intake however remains a controversial approach [49,50,51] and previous studies suggest neutral effects from increased *n*-6 intake to BP lowering [52,53]. As intake of essential *n*-6 linoleic acid (C18:3 *n*-6) and *n*-3 ALA are highly correlated since both are abundant in plant oils, observed inverse associations might reflect beneficial implications of higher intake of dietary ALA itself and its metabolic products [46,54]. 

Finally, our results should be placed in the context of a population under urbanisation coupled with transitions in nutritional habits. The protective 10-year associations of AA and osbond acid remained significant in urban dwellers only. In urban subjects only we observed decrease in EPA and DHA in plasma phospholipids, possibly be due to westernised dietary patterns characterized by cooking oils rich in linoleic acid and *n*-6 PUFA [55] and poor intake of *n*-3 sources (such as whole grains, vegetables and marine food). The finding on EPA and DHA decrease thus supports existing policies on increasing *n*-3 intake in this population undergoing urbanisation [56]. In rural subjects only we observed an increase in DHA, which was also associated with lower DBP. It is less plausible that the increase was due to pronounced intake of DHA from marine food, rather a consequence of enhanced conversion towards long-chain *n*-3 products within rural subjects with significantly lower *n*-3 intake. 

Lack of consistent association between plasma *n*-3 PUFA and BP in our study is partly in line with recent findings from ASCEND trial conducted in 16,000 diabetic middle-aged and older subjects [57]. The authors demonstrated no beneficial effects of daily consumption of *n*-3 fish oil capsules (460 mg EPA + 380 mg DHA) in comparison with placebo olive oil, and regarding incidence of serious vascular events upon 7.4 years follow-up [57]. On the other hand, REDUCE-IT showed that among 8000 patients with elevated triglycerides and stable LDL-cholesterol, receiving 2 g of highly purified EPA ethyl ester twice daily was associated with significantly lower risk of composite cardiovascular event, in comparison with placebo and despite the use of statins [58]. Based upon our results and considering the low *n*-3 PUFA intake [5] we may not discard the role of dietary *n*-3 PUFA and particularly EPA in strategies towards BP optimisation in Africa, and future intervention studies with increasing *n*-3 intake should elucidate the relationship.

The strength of our study lays in a repeated-measures design, evaluating time-dependent changes in BP and hypertension related to fat intake and metabolism. Furthermore, urbanisation-specific analyses and inclusion of a panel of demographic and clinical confounders provide robustness to the obtained relationships. We reported dietary and fat intake profiles across urbanisation categories in line with previously reported baseline dietary intake for the complete cohort (*n* 1950) [5] implying generalizability of our results to the population of black South Africans. Herein reported plasma fatty acid profiles are comparable to recent report within larger sample (*n* 711) [18] of the same cohort of black South Africans participating PURE, outweighing potential concern on the limited sample size of 300 subjects. Of note, 10-year attrition rate might have blurred some of the associations. However, we applied longitudinal analysis accounting for time-dependent variation of outcomes and exposures, providing additional reliability to the observed associations. We followed no changes in usage of medication or any other lifestyle confounder, potentially limiting our results. Although we accounted for an array of structured lifestyle, demographic and clinical confounders, the residual confounding cannot be ruled out.

In conclusion, our data advocate for a link between fat intake, blood pressure and urbanisation in a population of black South Africans with historically low omega-3 intake. Ten-year hypertension prevalence increased in the 300 subjects and only in urban residents did we observe a tendency towards 10-year optimization of hypertension status. Regardless of urbanisation areas there was an increase in individual plasma *n*-6 PUFA over 10 years, but only in urbans there was a decrease in EPA and DHA status, supporting policies on *n*-3 dietary reinforcement. The individual *n*-6 PUFA were inversely associated with blood pressure, prominently within urban dwellers. Taken together the results imply a protective mechanism linked with fat metabolism and vascular health in black South African population undergoing rapid nutritional transition. Indicated population-specific metabotype in black South Africans is possibly linked with genetic background and further research on FADS1 and FADS2 variants, desaturase activity and association with vascular function is warranted in the population.

## Figures and Tables

**Figure 1 foods-08-00394-f001:**
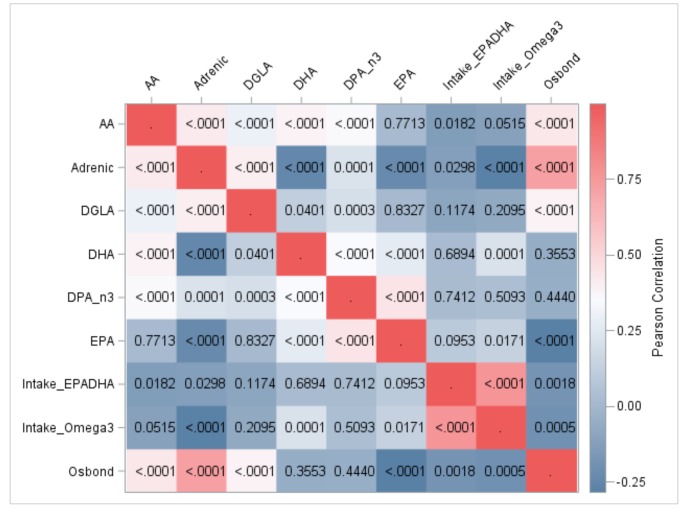
Baseline correlations between intake of *n*-3 fats and long-chain plasma fatty acids in 300 black South Africans: HeatMap of Pearson coefficients. DGLA, dihomo-γ-linoleic acid; AA, arachidonic acid; EPA, Eicosapentaenoic acid; DPA_n3, docosapentaenoic acid; DHA, Docosahexaenoic acid; Intake_EPADHA, Cumulative intake of preformed EPA and DHA; Intake_Omega3, Cumulative intake of EPA, DHA and plant-originated essential α-linolenic acid; <.0001, denotes statistical threshold (*p*) of < 0.0001 associated with correlation pair.

**Figure 2 foods-08-00394-f002:**
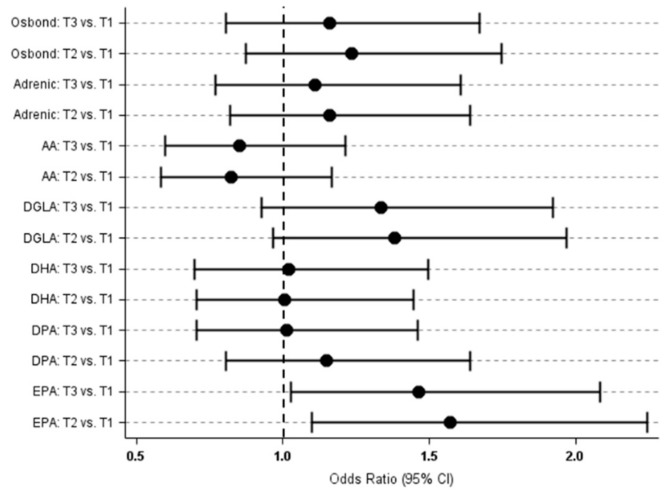
Multivariable odds ratio of being hypertensive across 10-years depending on the individual fatty acid content in plasma phospholipids in 300 black South Africans. AA, Arachidonic acid; DGLA, Dihomo-γ-linolenic acid, EPA, Eicosapentaenoic acid; DPA, docosapentaenoic acid; DHA, Docosahexaenoic acid; T1, T2, T3, Increasing tertiles of each plasma phospholipid fatty acid content.

**Table 1 foods-08-00394-t001:** Baseline characteristics of the 300 rural and urban black South Africans.

	Complete Sample (*n* 300)	Rural Areas (*n* 140)	Urban Areas (*n* 160)	*p*-Value ^1^
Gender, men, *n* (%)	91 (30.33)	39 (27.86)	52 (32.5)	0.385
Any education, *n* (%)	190 (64.63)	65 (47.10)	125 (80.13)	<0.001
Current smokers, *n* (%)	155 (51.84)	74 (52.86)	81 (50.94)	0.201
Former smokers, *n* (%)	135 (45.15)	59 (42.14)	76 (47.8)	
Clinical parameters				
Age, years	53.12 (9.83)	52.20 (9.16)	53.93 (10.34)	0.130
Body mass index, kg m^−2^	23.91 [19.93, 29.61]	23.39 [19.44, 29.43]	24.03 [20.43, 29.61]	0.172
Waist circumference. cm	80.68 (12.99)	79.84 (13.43)	81.42 (12.60)	0.295
Systolic blood pressure, mmHg	136.54 (3.29)	131.56 (22.81)	140.91 (22.89)	0.001
Diastolic blood pressure, mmHg	89.04 (12.69)	87.10 (13.33)	90.75 (11.87)	0.013
Fasting glucose, mmol L^−1^	5.11 (1.53)	4.87 (1.08)	5.32 (1.82)	0.010
Total cholesterol, mmol L^−1^	5.22 (1.30)	5.12 (1.32)	5.31 (1.27)	0.193
HDL-c, mmol L^−1^	1.58 (0.64)	1.55 (0.65)	1.62 (0.64)	0.338
LDL-c, mmol L^−1^	3.01 (1.18)	3.00 (1.16)	3.03 (1.19)	0.820
Tryglicerides, mmol L^−1^	1.15 [0.84, 1.68]	1.10 [0.84, 1.53]	1.23 [0.84, 1.75]	0.142
Weighted physical activity index	2.76 [2.49, 3.17]	3.07 [2.61, 3.43]	2.62 [2.38, 2.89]	<0.0001
GGT, U L^−1^	43.00 [29.00, 85.06]	37.30 [26.90, 64.35]	50.29 [34.38, 95.50]	0.001
hsCRP, mg L^−1^	3.19 [1.04, 7.52]	3.32 [0.90, 7.36]	2.90 [1.07, 8.39]	0.722
Use of hypertension medication, *n* (%)	55 (18.3)	27 (19.3)	28 (17.5)	0.691
Hypertensive, *n* (%)	176 (58.7)	68 (48.6)	108 (67.5)	<0.001
Dietary intake				
Energy, kJ	7251.15 [5259.26, 9689.23]	6103.36 [4681.22, 7928.72]	8453.99 [5824.48, 11439.46]	<0.0001
Total fat, g	43.14 [27.73, 63.05]	30.53 [21.94, 42.22]	59.04 [40.89, 82.64]	<0.0001
Saturated fat, g	9.97 [5.98, 16.29]	6.61 [3.88, 9.20]	15.18 [10.10, 21.39]	<0.0001
Monounsaturated fat, g	10.92 [6.10, 18.05]	6.79 [4.20, 10.00]	16.20 [11.14, 24.69]	<0.0001
Polyunsaturated fat, g	13.55 [7.85, 20.23]	9.59 [6.50, 14.26]	17.10 [11.49, 23.60]	<0.0001
*n*-3 intake, mg	314.10 [188.98, 476.17]	209.19 [137.58, 314.34]	425.59 [298.12, 608.72]	<0.0001
EPA+DHA intake, mg	109.91 [49.20, 199.49]	79.91 [34.66, 137.56]	130.48 [58.90, 230.20]	<0.0001
Total carbohydrate, g	279.98 (129.89)	256.77 (113.15)	300.29 (140.15)	0.003
Total fibre, g	21.35 (10.48)	18.62 (8.11)	23.75 (11.68)	<0.0001
Soluble fibre, g	1.40 [0.84, 2.32]	0.97 [0.66, 1.43]	2.12 [1.28, 3.32]	<0.0001
Alcohol, g	0.00 [0.00, 11.50]	0.00 [0.00, 5.71]	0.10 [0.00, 15.33]	0.010

HDL-c, High-density lipoprotein cholesterol; LDL-c, Low-density lipoprotein cholesterol; GGT, γ-glutamyl transferase; EPA, Eicosapentaenoic acid; DHA, Docosahexaenoic acid; *n*-3, Intake of EPA, DHA and plant-originated α-linolenic acid. Data are presented as mean (SD), median [25th, 75th] or percentage for categorical variables. ^1^ Significance values calculated by use of independent *t*-test or Mann-Whitney test.

**Table 2 foods-08-00394-t002:** Plasma phospholipid fatty acid status across 10 years in 300 black South Africans.

	Study Year			*p* ^1^
	2005	2010	2015	
Myristic acid, 14:0	0.27 (0.01)	0.27 (0.01)	0.33 (0.03)	<0.0001
Palmitic acid, 16:0	26.93 (0.31)	27.17 (0.45)	24.79 (0.64)	<0.0001
Palmitoleic acid, 16:1 *n*-7	0.86 [0.76, 0.96]	0.83 [0.70, 0.91]	0.93 [0.79, 1.04]	0.086
Stearic acid, 18:0	15.26 (0.96)	14.92 (0.61)	14.19 (0.07)	<0.0001
Oleic acid, 18:1 *n*-9	8.84 [8.35, 9.27]	8.48 [7.88, 8.95]	8.33 [7.73, 8.63]	0.025
Mead, 20:3 *n*-9	0.25 [0.25, 0.26]	0.24 [0.22, 0.25]	0.27 [0.19, 0.28]	0.216
Linoleic acid, 18:2 *n*-6	16.03 (0.41)	16.70 (1.01)	16.07 (0.28)	0.579
γ-Linolenic, 18:3 *n*-6	0.12 [0.11, 0.12]	0.12 [0.11, 0.13]	0.11 [0.10, 0.11]	0.018
Dihomo-γ-linolenic, 20:3 *n*-6	2.91 (0.09)	2.89 (0.08)	3.48 (0.19)	<0.0001
Arachidonic acid, 20:4 *n*-6	13.57 (0.24)	14.65 (0.31)	18.13 (0.37)	<0.0001
Adrenic, 22:4 *n*-6	0.60 (0.07)	0.70 (0.01)	0.66 (0.02)	<0.0001
Osbond, 22:5 *n*-6	0.57 [0.56, 0.67]	0.72 [0.70, 0.73]	1.07 [0.93, 1.09]	<0.0001
α-linolenic acid, 18:3 *n*-3	0.09 [0.09, 0.09]	0.09 [0.09, 0.11]	0.07 [0.07, 0.08]	<0.0001
EPA, 20:5 *n*-3	0.78 [0.59, 0.80]	0.47 [0.45, 0.60]	0.55 [0.52, 0.59]	<0.0001
Docosapentaenoic, 22:5 *n*-3	1.41 (0.02)	1.42 (0.19)	1.53 (0.08)	0.001
DHA, 22:6 *n*-3	4.56 (0.61)	3.88 (0.11)	4.33 (0.14)	0.009

EPA, Eicosapentaenoic acid; DHA, Docosahexaenoic acid. Age and urbanization factor-adjusted data presented as mean (SD) or median [25th, 75th]. ^1^ Probability trends associated with changes over 10-years calculated by general linear model adjusted for age and urbanization factor.

**Table 3 foods-08-00394-t003:** Ten-year associations between plasma phospholipid long-chain fatty acids and blood pressure in 300 black South Africans.

	Systolic Blood Pressure		Diastolic Blood Pressure	
	β (95% CI)	*p* ^3^	β (95% CI)	*p* ^3^
Long-chain *n*-3 fatty acids				
EPA, 20:5 *n*-3				
T1	ref.		ref.	
T2^1^	1.89 (−1.40, 5.18)	0.322	1.44 (−0.44, 3.32)	0.143
T3	2.41 (−0.89, 5.70)		1.80 (−0.09, 3.69)	
T2^2^	2.39 (−0.95, 5.72)	0.191	1.37 (−0.56, 3.30)	0.132
T3	2.92 (−0.41, 6.26)		1.94 (0.01, 3.87)	
Docosapentaenoic, 22:5 *n*-3				
T1	ref.		ref.	
T2 ^1^	0.91 (−2.39, 4.22)	0.284	0.58 (−1.31, 2.48)	0.056
T3	−1.75 (−5.26, 1.75)		−1.69 (−3.69, 0.30)	
T2 ^2^	0.22 (−3.10, 3.53)	0.349	0.33 (−1.59, 2.24)	0.068
T3	−2.11 (−5.63, 1.41)		−1.86 (−3.88, 0.17)	
DHA, 22:6 *n*-3				
T1	ref.		ref.	
T2 ^1^	−0.92 (−4.31, 2.47)	0.386	−0.76 (−2.70, 1.18)	0.275
T3	−2.48 (−6.08, 1.11)		−1.68 (−3.73, 0.38)	
T2 ^2^	−0.37 (−3.79, 3.06)	0.427	−0.51 (−2.49, 1.46)	0.358
T3	−2.21 (−5.86, 1.44)		−1.50 (−3.60, 0.60)	
Long-chain *n*-6 fatty acids				
Dihomo-γ-linolenic acid, 20:3 *n*-6				
T1	ref.		ref.	
T2 ^1^	1.40 (−1.93, 4.73)	0.419	0.55 (−1.36, 2.46)	0.396
T3	−0.77 (−4.17, 2.63)		−0.75 (−2.70, 1.19)	
T2 ^2^	0.68 (−2.70, 4.06)	0.392	0.23 (−1.72, 2.19)	0.305
T3	−1.59 (−5.07, 1.89)		−1.19 (−3.20, 0.82)	
Arachidonic acid, 20:4 *n*-6				
T1	ref.		ref.	
T2 ^1^	−0.06 (−3.29, 3.16)	0.048	−0.83 (−2.66, 1.00)	<0.0001
T3	−3.50 (−6.73, −0.27)		−3.76 (−5.59, −1.93)	
T2 ^2^	0.17 (−3.06, 3.39)	0.024	−0.62 (−2.47, 1.22)	<0.0001
T3	−3.81 (−7.07, −0.54)		−3.82 (−5.70, −1.95)	
Adrenic acid, 22:4 *n*-6				
T1	ref.		ref.	
T2 ^1^	−1.87 (−5.11, 1.37)	0.327	0.03 (−1.83, 1.88)	0.999
T3	0.45 (−3.02, 3.92)		−0.02 (−2.00, 1.97)	
T2 ^2^	−2.56 (−5.79, 0.68)	0.195	−0.23 (−2.10, 1.65)	0.943
T3	0.00 (−3.52, 3.53)		0.09 (−1.94, 2.13)	
Osbond acid, 22:5 *n*-6				
T1	ref.		ref.	
T2 ^1^	−1.74 (−4.97, 1.48)	0.449	−1.22 (−3.05, 0.61)	0.002
T3	−2.03 (−5.51, 1.45)		−3.47 (−5.44, −1.49)	
T2 ^2^	−1.96 (−5.18, 1.26)	0.197	−1.22 (−3.07, 0.63)	0.001
T3	−3.20 (−6.73, 0.33)		−3.71 (−5.73, −1.70)	

EPA, Eicosapentaenoic acid; DHA, Docosahexaenoic acid; T1, T2, T3, Increasing tertiles of plasma phospholipid fatty acid content. ^1^ Model 1 adjusted for age, gender and urbanization factor. ^2^ Model 2 further adjusted for level of education, use of tobacco, use of hypertension medication, body mass index, physical activity index and dietary intake of alcohol (g). ^3^ Probability values associated with β estimating absolute change in blood pressure (in mmHg) with regards to 10-year change in a fatty acid level.

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
