# Peer review of "Long-Chain Polyunsaturated Fatty Acids Are Associated with Blood Pressure and Hypertension over 10-Years in Black South African Adults Undergoing Nutritional Transition"

_foods, 2019, doi:10.3390/foods8090394_

Round 1

Reviewer 1 Report

I enjoyed reading your interesting manuscript. Please consider the following:

Abstract:

-I think that the absolute changes in blood pressure would be a more appropriate value to include in the abstract than the odds ratio if hypertension. The absolute associations were relatively small +2.9 mmHg (SBP) and +1.94 (DBP), but because the starting BP was high (137/89) and given that BP is expected to increase over time, these small increases resulted in a number of individuals exceeding the arbitrary 140/90 threshold resulting in the 1.46 higher odds of hypertension. Without the context of the absolute values, this OR might be misinterpreted - I think further elaboration of this is needed in the discussion.

Methods:

-Please state when the FFQ was applied

-Please state whether either or both SBP and DBP had to exceed thresholds for a diagnosis of hypertension to be made.

Results

-Please consider adding more details to table legend 3 - explain that these figures are mmHG, not ORs.

Discussion

-Please discuss the potential consequences of attrition bias on your findings

-Please consider citing and discussing the REDUCE-IT trial https://www.nejm.org/doi/full/10.1056/NEJMoa1812792

-Please consider the potential consequences of residual confounding on your findings

Author Response

Dear Reviewer 1,

Thank you for you contribution. Attached please find a file containing point-wise comments and suggestions addressed.

Kind regards

Reviewer 2 Report

foods-585654

Good and well written paper reporting counterintuitive results, i.e. EPA associated with higher BP despite what Dariush Mozaffarian says and AA associated with lower BP despite what Joe Hibbeln says.

These results are likely to stir some controversy and the authors might want to make it bullet-proof.

One way it to criticize the PURE study. This study gave the wrong impression that carb consumption is associated with higher mortality. However, the investigators reported consumption as percentage, showing that if you only eat white rice your prognosis is poorer than if you add some fat and protein. Big surprise. In this respect, food transition in South African rural areas should be controlled yet encouraged and lay press slogans that eating more fish (=EPA) will give you hypertension are around the corner. The part of the Discussion on racial metabolic differences is very interesting and warranted and, as mentioned, the lack of genetic profiling, namely FADS is one of the major limitations of this paper.

Do the authors plan a follow up?

Finally, add a few words on guidelines, including the [outdated] ISSFAL one, the AHA one, etc.

Author Response

Dear Reviewer 2,

Thank you for you contribution. Attached please find a file containing point-wise comments and suggestions addressed.

Kind regards
